# CIRCUIT: A BENCHMARK FOR CIRCUIT INTERPRETATION AND REASONING CAPABILITIES OF LLMS

## ABSTRACT

The role of Large Language Models (LLMs) has not been extensively explored in analog circuit design, which could benefit from a reasoning-based approach that transcends traditional optimization techniques. In particular, despite their growing relevance, there are no benchmarks to assess LLMs' reasoning capability about circuits. Therefore, we created the CIRCUIT dataset consisting of 510 question-answer pairs spanning various levels of analog-circuit-related subjects. The best-performing model on our dataset, GPT-4o, achieves 48.04% accuracy when evaluated on the final numerical answer. To evaluate the robustness of LLMs on our dataset, we introduced a unique feature that enables unit-test-like evaluation by grouping questions into unit tests. In this case, GPT-4o can only pass 27.45% of the unit tests, highlighting that the most advanced LLMs still struggle with understanding circuits, which requires multi-level reasoning, particularly when involving circuit topologies. This circuit-specific benchmark highlights LLMs' limitations, offering valuable insights for advancing their application in analog integrated circuit design.

## 1 INTRODUCTION

The application of Large Language Models (LLMs) in analog integrated circuit design could pioneer a new era of AI applications in domains traditionally dominated by human expertise. Analog semiconductor chips are the core building blocks in sensing and communication systems. Contrary to digital chip development, where computer-aided design automation has been widely adopted for a few decades, analog design, often perceived more as a craftsmanship than a well-established engineering procedure, relies heavily on the designer's experience and intuition to navigate in the trade space of efficiency, noise, linearity, and speed to meet certain specifications. This domain's depth, requiring a blend of acumen and creativity, underscores the high barriers to entry and the extensive training required to master its intricacies, which exacerbated the critical labor shortfall of the semiconductor industry in this decade (Ravi, 2023).

The advent of AI-assisted design automation in analog circuit design holds considerable promise to tackle the aforementioned challenge. It offers the potential to significantly streamline design cycles, enabling engineers to focus more on strategic, high-level design considerations and the exploration of novel ideas and applications. Traditional analog design automation (Wang et al., 2018; Settaluri et al., 2020; Liu et al., 2022; Xue et al., 2023; Zhang et al., 2019) has relied on numerical-based optimization and machine learning techniques to train surrogate models for designing circuits with fixed topologies and semiconductor processes, resulting in reduced generalization capabilities and often suffering from limited interpretability. A shift towards a reasoning and knowledge-based approach, facilitated by LLMs that transcend traditional optimization techniques, could leverage circuit domain expertise to innovate and refine the design of diverse analog circuits.

A natural starting point towards this ambitious goal is to evaluate existing LLMs' proficiency in executing various analog circuit design tasks. To that end, we introduce the **CIRCUIT** (Circuit Interpretation and Reasoning Capabilities) dataset , which focuses on simple topology understanding – a precursor to performing any complex design task. The dataset is designed to be scalable, enabling a seamless incorporation of more advanced analog circuit design tasks in future iterations. We evaluate leading LLMs' performance on the dataset with a unique, template-based evaluation metric. Furthermore, we conduct automatic and human evaluation and error analysis of the LLM responses.

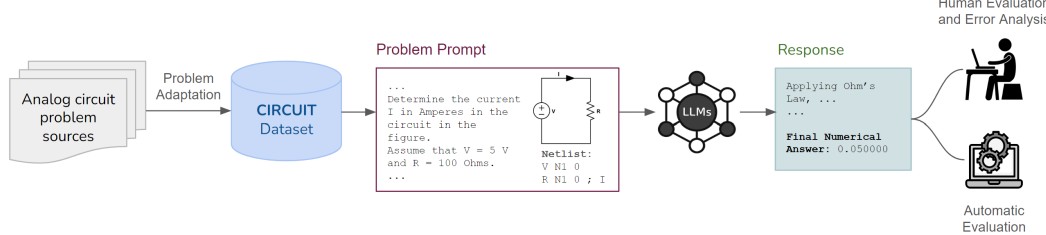

Figure 1: A simplified overview of the CIRCUIT dataset and experiment setup. Analog circuit problems, sourced from various materials, are adapted into the CIRCUIT dataset, comprising 510 problems. We assess the performance of three Large Language Models (GPT 4o, GPT Turbo, Gemini 1.5 Pro) in understanding analog circuits and their topologies from diagrams and netlists, using four distinct prompt designs. The LLMs' responses are then evaluated both automatically and manually, with unique evaluation metrics designed to reveal higher-level insights and capture the effects of data homogeneity. Quantitative analysis and human error analysis were done to assess model performance in reasoning about analog circuits.

## 2 RELATED WORK

Task-specific evaluation plays a crucial role in advancing research in LLM applications by providing precise insights into model capabilities and limitations within defined contexts. The scalability of general-purpose models has demonstrated enhanced task performance in various domains, including language (Brown et al., 2020), mathematics (Aojun Zhou, 2023; Mao et al., 2024), and code generation (Chen et al., 2021)[1].

In the realm of digital circuit design, noteworthy progress has been made in harnessing LLMs for tasks such as generating Verilog Code, as explored by Mingjie Liu (2023). Moreover, Cadence's JedAI [2] platform exemplifies the first application of LLM technology in chip design, illustrating the feasibility of integrating LLMs into digital design workflows.

In the realm of analog design, LLMs have already been integrated into frameworks that automate aspects of the design process (Chang et al., 2024; Lai et al., 2024). While these works focus on leveraging LLMs directly for circuit design, an essential precursor is to evaluate the knowledge and reasoning capabilities of LLMs on fundamental analog circuit knowledge. Without a deep understanding of their foundational capabilities, the effectiveness and versatility of LLMs in real-world circuit design may be limited. To address this gap, we introduce the **CIRCUIT** dataset, which serves as a critical first step in the analog design pipeline.

When reviewing existing datasets for other domains, we notice that evaluation proves difficult on complex tasks. Coding tasks utilize unit testing with automatic evaluation, while other fields necessitate human evaluation. LLMs have also been used as evaluating agents. (Mao et al., 2024; Lin et al., 2021) While LLMs can evaluate large volumes of data, do not suffer from fatigue, and are cheaper to utilize, our initial experiments showed that they struggle with understanding and interpreting complex reasoning about analog circuits. Inspired by unit testing, we introduce a simple dataset design and evaluation metric combination that shows promise for the assessment of LLMs across various fields and tasks. This framework is inherently scalable, adaptable to more complex analog design tasks, and transferable to other reasoning domains.

---

[1]HumanEval

[2]JedAI

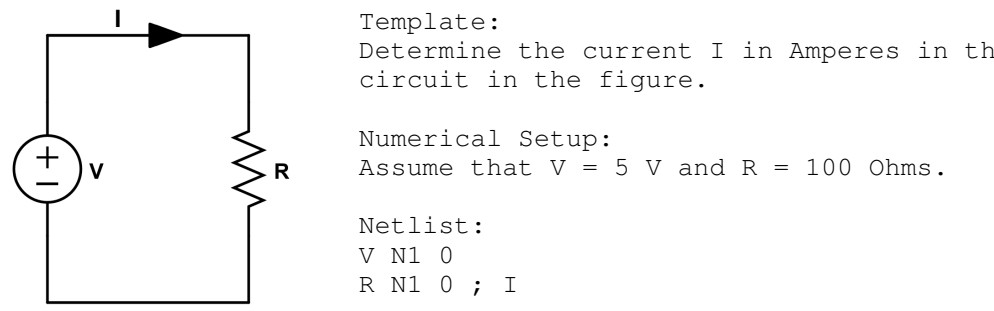

```
Template:
Determine the current I in Amperes in the
circuit in the figure.

Numerical Setup:
Assume that V = 5 V and R = 100 Ohms.

Netlist:
V N1 0
R N1 0 ; I
```

Figure 2: Example datapoint from the CIRCUIT dataset. Each datapoint includes a template question, which may or may not have an associated diagram. In most cases, diagrams are further supplemented by netlists that describe the circuit's components and connections. Additionally, each datapoint is associated with a unique numerical setup.

# 3 CIRCUIT DATASET

## 3.1 DATASET CURATION

The **CIRCUIT** dataset comprises circuit problems, many of which include associated diagrams. The dataset was made using templates – problems adapted from sources listed in Appendix A modified to fit different numerical setups and ensure each only asks for a single numerical answer. Figure 2 is an example of a dataset question. The diagram and the template are adapted so that the numerical setup can accommodate different values and ensure different answers to the template question. Therefore, we were able to create multiple numerical setups for each template used for the creation of the dataset. Each template question together with its numerical setups served as a single unit test in the dataset. This design enables a more nuanced evaluation of the models' understanding of different circuit topologies and provides quantifiable insights into how data homogeneity influences model performance.

Initial experiments indicated that LLMs found it challenging to interpret circuit diagrams, particularly the direction and orientation of circuit components. To aid in understanding circuit topologies, we incorporated netlists into the prompts. Netlist syntax was slightly modified to better suit our needs, detailed in Appendix B. This modification and the inclusion of a syntax explanation in the prompts were aimed at enhancing LLMs' performance on our dataset.

Figure 2 illustrates an example of a data point consisting of a template question along with its associated diagram, netlist, and a numerical setup. In this scenario, the LLM is tasked with applying Ohm's law ($V = IR$) to calculate the current. The specific setup prompts for a calculation of $I = \frac{V}{R} = \frac{5V}{100\Omega} = 0.05A$, testing the LLM's understanding of this simple circuit topology. Our dataset extends this approach by using various values for $V$ and $R$ for numerical setups, thus methodically exploring the output curve $I$ in a unit-test-like fashion. That is, to test the understanding of this topology, we create multiple data points with different numerical setups, each maintaining the same structure, template question, diagram, and netlist but altering $V$ and $R$ values in the numerical setup to produce data points with different correct answers. Providing correct answers to each numerical setup strongly suggests an understanding of the topology, without requiring a detailed examination of the solution methodology, much like how unit tests in programming verify that a function is correctly implemented.

## 3.2 DATASET STATISTICS

The **CIRCUIT** dataset consists of 510 questions derived from 102 templates, with 5 numerical setups each. 93 templates include diagrams, 79 of which include netlists. Templates are divided into four categories—basic, analog, power, and radio-frequency (RF)—and are graded by levels based on the corresponding MIT course and the typical class year. For example, MIT 6.002 (Circuits and

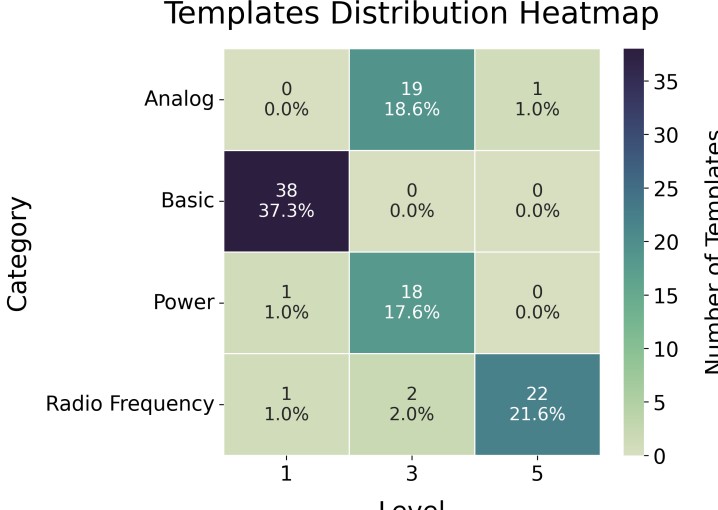

Figure 3: Templates distribution across categories and levels. The heatmap displays the distribution of templates in the CIRCUIT dataset across four categories (Analog, Basic, Power, and Radio Frequency) and three levels (1, 3, and 5). The numbers inside each cell represent the total count of templates, with percentages indicating the proportion of templates relative to the entire dataset (totaling 102 templates). The color intensity corresponds to the number of templates, as indicated by the color bar on the right.

Electronics) problems are level 1 since the class is typically taken by freshmen. The category-level distribution of the dataset is given in Figure 3.

## 4 EVALUATION

### 4.1 METRICS

As previously described, each template $t_i$ is associated with $n = 5$ distinct numerical setups in the dataset. These setups yield straightforward numerical outcomes and aim to cover the comprehensive output range pertinent to the respective circuit.

We evaluate using both global and template-level accuracies. Global accuracy is defined as:

$$A_{\text{global}} = \frac{\text{\# correctly answered questions}}{\text{\# total questions}}$$

for the entire dataset and its subsets.

Template accuracy, which leverages the unit-test-like structure of our dataset, is gauged by the **pass@k/n** metric. This metric evaluates the model's understanding of a single circuit topology through $n$ numerical setups ($n = 5$ for our dataset), which make up a unit test. A template is considered accurate (i.e. a unit test is passed) if at least $k$ of its $n$ setups are correctly solved. Therefore, the template accuracy is defined as:

$$A_{\text{template},k/n} = \frac{\sum_{i=1}^{m} A_{t_i,k}}{m}, \quad \text{where} \quad A_{t_i,k} = \begin{cases} 1 & \text{if at least } k \text{ out of } n \text{ setups} \\ & \text{are answered correctly} \\ 0 & \text{otherwise} \end{cases}$$

and reported for various values of $k$ across all 102 templates ($m = 102$) and their subsets.

### 4.2 METHODS

Our straightforward numerical setups allow for the automatic evaluation of LLM performance. We prompt LLMs to give their final numerical answers in a specified format (details in Appendix C)

and facilitate parsing via regex from the responses. Additionally, we conduct human evaluations on a subset of responses for error and qualitative analyses.

## 5 EXPERIMENTS

### 5.1 MODELS

We evaluated `gpt-4-turbo` (OpenAI, 2024), `gpt-4o` (OpenAI, 2024) and `gemini-1.5-pro` (Team et al., 2024) on our dataset, setting the maximum tokens to 1,536 for each. Detailed prompt design is available in Appendix C. Following well-established prompting techniques (Brown et al., 2020; Schulhoff et al., 2024), four different prompts were tested for each model: zero-shot and one-shot, with and without netlists. Models were instructed to give their final numerical answers with a precision of six decimal places.

### 5.2 EXPERIMENTS

In each experiment, models were provided with diagrams for questions that included them. In the first 3 experiments, models received all questions from the CIRCUIT dataset with a 0-shot prompt. In the next 3, models were given 395 questions that had associated netlists and the same 0-shot prompt, along with netlists and customized instructions for interpreting only the elements present in each netlist. In the third set of 3 experiments, models were given all questions with a 1-shot prompt. In the final 3 experiments, models received all questions, a 1-shot prompt with a netlist example, netlists, and the necessary netlist instructions. Details of the prompt design are in Appendix C. Responses from all experiments were quantitatively analyzed, with a subset reviewed for errors and qualitative insights by human evaluators.

### 5.3 EVALUATION

We used an automatic evaluation method to assess model responses and reported both global and template accuracies. Responses were deemed correct if the absolute difference from the ground truth was less than 0.001. Additionally, we conducted a human evaluation of best-accuracy responses to verify automatic evaluation results, analyze errors, and understand the qualitative aspects of the responses. Errors were categorized into mathematical, response formatting, and reasoning. The models sometimes displayed clear misunderstandings of the circuit topology, which we classified as topology errors, a specific type of reasoning error. A common topology error was misunderstanding element orientation or direction, the rate of which we also reported. More details on error types and subtypes can be found in Appendix D. Human evaluation deemed responses as correct if they were devoid of errors.

## 6 RESULTS

### 6.1 QUANTITATIVE ANALYSIS

#### 6.1.1 AUTOMATIC EVALUATION

We assessed model performance across the entire CIRCUIT dataset using automatic evaluation, with results detailed in Table 1. A key observation is that the best-performing prompt varies by model and by the specific accuracy metric. For instance, `GPT 4-turbo` achieves the highest global accuracy with the 1-shot prompt, while its highest 5/5 template accuracy occurs with the 1-shot prompt with a netlist example. In contrast, `Gemini 1.5-pro` performs best with the plain 0-shot prompt across all metrics, indicating a potential struggle to integrate additional information from netlists or example-based problem-solving strategies provided in the 1-shot prompts. The most consistent and highest-performing model across both global and template accuracies appears to be `GPT 4o`, which leverages netlists effectively but does not seem to gain further advantage from the 1-shot prompt.

One important pattern we observe is that template accuracy decreases as the value of $k$ in pass@k/n increases. This reflects the increasing difficulty in achieving correctness across all five numerical

setups in a given template. Notice that pass@3/5 template accuracy closely aligns with global accuracy indicating that relying solely on global accuracy can obscure deeper insights into a model's performance on the given dataset.

Table 2 provides further granularity by separating results into two subsets: questions with and without associated netlists. `GPT 4o` outperforms other models in both subsets. Notably, questions without netlists yield higher average scores, likely due to their emphasis on reasoning which does not require the model to understand complex circuit topologies. All models benefit from the 1-shot example in this subset, with `GPT 4-turbo` showing the most significant improvement when the netlist is included in the 1-shot example. For questions with netlists, model preferences diverge. While `GPT 4o` performs best with the 0-shot prompt including netlists, its template accuracy for higher values of $k$ remains strong even with the 1-shot prompt including netlists. `Gemini 1.5-pro` does not seem to benefit from additional information in the prompts, and `GPT 4-turbo` shows mixed results between global and template metrics.

The global accuracies indicate that, despite the complexity and the specialized knowledge required for the CIRCUIT dataset, the models show reasonable performance. However, the template accuracies reveal that the range of circuit topologies the models can grasp is limited.

### 6.1.2 HUMAN EVALUATION

Automatic evaluation predominantly assesses model outputs by comparing them to numeric ground truths and typically does not penalize incorrect reasoning. Concerns about this method also include mathematical errors and incorrect response formatting. `GPT 4o` was selected for a detailed human evaluation because it demonstrated superior performance in the automatic assessment.

Results outlined in Table 3 affirm that the trends observed in human evaluations are consistent with those from automatic evaluation. To further understand the correlation between automatic and human evaluations, we analyzed the occurrence of false positives—instances where responses were deemed correct by automatic metrics but identified as incorrect upon human review. Approximately 5% of the automated evaluations resulted in false positives, impacting even the most rigorous template accuracies. Despite these occasional discrepancies, automatic evaluation proves to be a dependable tool for understanding model performance.

Human evaluation involved a thorough error analysis, detailed in Table 4, with error categorization methodologies explained in Appendix D. The primary error types identified were mathematical, formatting, and reasoning—the latter encompassing all errors not directly related to mathematical or formatting issues. Within reasoning errors, misunderstandings related to topology emerged as a significant subcategory, and issues with direction or orientation of elements were recognized as a specific concern within topology errors. Our analysis indicates that mathematical and formatting errors constitute a minor portion of the total errors, and the predominant challenges for models stem from reasoning errors. This highlights the complexity of our dataset which requires a deep understanding of underlying concepts and their applications.

Additionally, global per-category and per-level accuracies on human-evaluated responses are summarized in Table 5 and Table 6 respectively. These results highlight the challenges in understanding more complex topologies, as evidenced by significantly lower performance on questions with netlists and at higher levels. Furthermore, the consistently higher accuracy in the 'Basic' category across configurations suggests that GPT-4o is better equipped to handle introductory-level circuits than more advanced ones.

### 6.2 QUALITATIVE ANALYSIS

GPT 4o's responses revealed that the model generally employed appropriate tools and formulas and understood which elements were present in the given circuit. However, it struggled with complex circuit topologies; even with netlists, higher-level reasoning remained challenging. Sometimes, even when given a netlist, GPT's response would not indicate its use. We also noticed that netlists often helped GPT understand a part of or the entire given topology. Errors often stemmed from misconceptions about interactions and connections between components and subcircuits. GPT also struggled with directions and element orientations, such as current flow direction from a current source. Sometimes, GPT made minor reasoning errors which didn't affect the correctness of the

final solution. While GPT occasionally made mathematical errors, these were primarily confined to approximation errors, often division and logarithmic and exponential calculations, and sometimes careless mistakes in equation manipulation, reinforcing that the primary challenge lies in reasoning rather than basic mathematics. Nevertheless, the fixed error on the final numerical answer was sometimes too stringent for GPT's approximations. GPT occasionally displayed conceptual misunderstandings, failed to follow given instructions, or applied general knowledge without adapting to specific contexts. Hallucinations about non-existent configurations were also noted. For instance, when given an op-amp in negative feedback, GPT hallucinated its non-inverting input was grounded.

This qualitative analysis underlines the nuanced challenges GPT faces with our dataset and gives us a glimpse into the data GPT was trained on. More specific examples can be found in Appendix E.

## 7 DISCUSSION & LIMITATIONS

### 7.1 DISCUSSION

Through our experiments, we gained valuable insight into the capabilities of existing LLMs in understanding and reasoning about various analog circuit topologies. Our quantitative and qualitative analyses indicate that these models possess reasoning abilities and relevant expert knowledge to tackle the problems in our dataset. Their understanding of circuit topologies can be improved when netlists and 1-shot examples are provided, but substantial work remains to be done to improve their performance further on our dataset. Addressing these basic shortcomings in topology understanding is crucial before advancing to more complex analog design tasks—both of which represent exciting directions for future work.

Our dataset design together with the pass@k/n metric enables an automatic evaluation framework for quick, cost-effective, reliable, and comprehensive automatic evaluation of LLMs' capabilities. pass@k/n offers a more nuanced understanding of model performance than a mere global accuracy score. On our dataset, it reveals that these models are proficient in only a narrow subset of topologies, and a closer look found this subset focused on very simple topologies. Enhancing this pass@k/n's potential to yield deeper insights into model understanding could be explored in future work by enriching templates with more detailed annotations. Uniquely, the metric can be adjusted for different levels of strictness (k), allowing researchers to evaluate model performance under varying levels of precision. The unit-test-like pass@k/n metric can be beneficial in domains beyond analog circuits where a deep understanding of nuanced subject matter is critical, and where datasets can be structured with multiple subcomponents per main category to assess comprehensive knowledge. Future work could investigate applying our dataset design and metric to new domains, different unit test designs for distinct evaluation goals, and strategies for evaluating intermediate steps in LLM reasoning to enable a more detailed assessment.

A key aspect of the CIRCUIT dataset design is its transparency regarding data homogeneity achieved through our unit test setup. When we compare global accuracy to template accuracy, we see the potential pitfall of relying solely on global metrics in dataset evaluation. Global accuracy provides an aggregate view of model performance but can mask nuanced failures that become apparent when assessing models on a template level. The CIRCUIT dataset's explicit design allows us to observe this distinction more clearly, as it isolates a model's ability to handle both the homogeneity (consistent core structures) and variability (changing numerical setups) inherent in real-world problems. This approach contrasts with traditional datasets, where either the homogeneity may not be explicit or the variability across problem instances may not be systematically controlled. By designing datasets like CIRCUIT, where the relationship between template structure and numerical variability is clear, we can gain deeper insights into model robustness and generalization capabilities. Template pass@k/n accuracies on our dataset show low generalization capabilities across variability in numerical setups. This is concerning for analog circuit design because it suggests that models struggle to adapt to different component values and configurations, which are critical for reliable performance in real-world circuit applications. Therefore, we encourage making homogeneity a more explicit aspect of dataset design and look forward to the insights future work may uncover.

Error analysis showed that most incorrect responses stemmed from reasoning errors, while mathematical inaccuracies were rare. There is a potential role for integrating a Python interpreter to mitigate mathematical errors, as noted by Gao et al. (2023). Qualitative analysis further revealed the

Table 1: **Accuracies for the Entire CIRCUIT Dataset.** The CIRCUIT dataset comprises 102 templates. **Accuracy is reported using two metrics: global (Glob.)**, which measures performance across the entire dataset, **and template**, which measures performance based on the smallest number of correct numerical setups per template (5/5, 4/5, and 3/5). The highest accuracies are bolded, and the best-performing prompt highlighted in green. The table presents the performance of three models (GPT-4-turbo, GPT-4o, and Gemini 1.5-pro) across various **prompt configurations: 0-shot (0-s), 0-shot with netlists and instructions (0-s w/ netlists) where applicable, 1-shot (1-s), and 1-shot with netlists and instructions (1-s w/ netlists).**

| Model | Prompt | Accuracies (%) | | | |
|---|---|---|---|---|---|
| | | Entire dataset (102 templates) | | | |
| | | Glob. | Template | | |
| | | | 5/5 | 4/5 | 3/5 |
| GPT 4-turbo | 0-s | 38.4 | 18.6 | 30.4 | 40.2 |
| | 0-s w/ netlists | 38.2 | 19.6 | 32.4 | 35.3 |
| | 1-s | 39.2 | 15.7 | 32.4 | 40.2 |
| | 1-s w/ netlists | 38.2 | 22.6 | 31.4 | 34.3 |
| GPT 4o | 0-s | 46.7 | **27.5** | 35.3 | **48.0** |
| | 0-s w/ netlists | **48.0** | **27.5** | **37.3** | 47.1 |
| | 1-s | 39.6 | 23.5 | 33.3 | 38.2 |
| | 1-s w/ netlists | 43.1 | 24.5 | 34.3 | 43.1 |
| Gemini 1.5-pro | 0-s | 36.3 | 18.6 | 29.4 | 33.3 |
| | 0-s w/ netlists | 34.7 | 13.7 | 25.5 | 33.3 |
| | 1-s | 32.0 | 10.8 | 21.6 | 30.4 |
| | 1-s w/ netlists | 32.2 | 13.8 | 23.5 | 33.3 |

Table 2: **Accuracies on CIRCUIT Dataset Subsets: questions which have associated netlists (Questions With Netlists) and questions which do not have associated netlists (Questions Without Netlists).** The table presents the performance of three models across various prompt configurations and accuracy metrics, as described in Table 1, for the two subsets of the dataset. Note that out of 102 templates in the dataset, 23 templates do not have associated netlists, while 79 templates do.

| Model | Prompt | Accuracies (%) | | | | | | | |
|---|---|---|---|---|---|---|---|---|---|
| | | Questions Without Netlists | | | | Questions With Netlists | | | |
| | | Glob. | Template | | | Glob. | Template | | |
| | | | 5/5 | 4/5 | 3/5 | | 5/5 | 4/5 | 3/5 |
| GPT 4-turbo | 0-s | 61.7 | 39.1 | 56.5 | 60.9 | 31.7 | 12.7 | 22.8 | 34.2 |
| | 0-s w/ netlists | | | | | 31.4 | 13.9 | 25.3 | 27.9 |
| | 1-s | 62.6 | 39.1 | 56.5 | 69.6 | 32.4 | 8.9 | 25.3 | 31.7 |
| | 1-s w/ netlists | 63.5 | 43.5 | 60.9 | 65.2 | 30.9 | 16.5 | 22.8 | 25.3 |
| GPT 4o | 0-s | 67.0 | 47.8 | **65.2** | **69.6** | 40.8 | **21.5** | 26.6 | **41.8** |
| | 0-s w/ netlists | | | | | **42.5** | **21.5** | **29.1** | 40.5 |
| | 1-s | **67.8** | **56.5** | **65.2** | 65.2 | 31.4 | 13.9 | 24.1 | 30.4 |
| | 1-s w/ netlists | 63.5 | 34.8 | 52.2 | **69.6** | 37.2 | **21.5** | **29.1** | 35.4 |
| Gemini 1.5-pro | 0-s | 55.7 | 26.1 | 56.5 | 56.5 | 30.6 | 16.5 | 21.5 | 26.6 |
| | 0-s w/ netlists | | | | | 28.6 | 10.1 | 16.5 | 26.6 |
| | 1-s | 56.5 | 26.1 | 43.5 | 65.2 | 24.8 | 6.3 | 15.2 | 20.3 |
| | 1-s w/ netlists | 53.0 | 21.7 | 43.5 | 56.5 | 26.1 | 11.4 | 17.7 | 26.6 |

Table 3: The table shows **the accuracy of GPT-4o responses evaluated automatically versus by humans**, using the metrics described in Table 1. It presents results for two prompts—0-shot with netlists and instructions, and 1-shot with netlists and instructions—on the **subset of the dataset with associated netlists (Questions With Netlist – 79 templates)**. Additionally, it includes results for the 1-shot prompt on the **subset without associated netlists (Questions W/O Netlists – 23 templates)**. The response subsets selected for human evaluation were chosen based on the results from Table 2.

| Dataset Subset | Prompt | GPT 4o Response Accuracies (%) | | | | | | | |
| --- | --- | --- | --- | --- | --- | --- | --- | --- | --- |
| | | Automatic | | | | Human | | | |
| | | Glob. | Template | | | Glob. | Template | | |
| | | | 5/5 | 4/5 | 3/5 | | 5/5 | 4/5 | 3/5 |
| Questions With Netlists | 0-s w/ netlists | 42.5 | 21.5 | 29.1 | 40.5 | 36.5 | 17.7 | 27.9 | 35.4 |
| | 1-s w/ netlists | 37.2 | 21.5 | 29.1 | 35.4 | 31.9 | 19.0 | 27.9 | 31.7 |
| Questions W/O Netlists | 1-s | 67.8 | 56.5 | 65.2 | 65.2 | 63.5 | 52.2 | 65.2 | 65.2 |

nature of the reasoning errors, pointing towards significant opportunities for improving the interpretative and reasoning capabilities of these models in future work.

Although the slight improvement in model accuracy with netlists suggests some sensitivity to additional contextual information, the overall impact remains modest. Interestingly, 1-shot prompting improved accuracy mainly on questions without associated netlists. The benefit of the 1-shot example isn't fully realized for questions involving netlists, possibly because the model sometimes fails to explicitly utilize the given netlist in its reasoning. Future work should explore the integration of more detailed contextual aids.

## 7.2 LIMITATIONS

This study, while insightful, faces several key limitations. The dataset's size and imbalance across categories, levels, and netlist presence could affect the generalizability of our findings, highlighting the need for a more representative dataset through expansion, particularly the number of numerical setups and better balancing. The dataset could be further enhanced by incorporating more challenging problems that better reflect contemporary circuit topologies. Additionally, the limited model selection and narrow focus in human evaluation limits our understanding of broader model capabilities.

## 8 CONCLUSION

We introduced **CIRCUIT**, the pioneering dataset designed specifically for assessing LLMs in the domain of analog circuit interpretation and reasoning. This work not only demonstrated the utility of meticulous and homogeneity-transparent dataset design but also highlighted the nuanced capabilities and limitations of leading LLMs through a comprehensive set of evaluations. The development of the pass@k/n metric and the strategic use of netlists significantly advanced our understanding of how models handle complex circuit topologies. Looking ahead, we encourage addressing the challenges posed by our dataset, expanding its scope, exploring our dataset design and metrics in other challenging domains, and further refining and developing our methodologies.

Table 4: **Human Error Analysis of GPT-4o Responses.** The table presents the error rates across different **error types (Math, Formatting, Reasoning, Topology, and Direction)** for GPT-4o responses analyzed by humans. **Error rates are calculated as the ratio of data points with the specified error to the total data points per subset**. It presents results for two prompts—0-shot with netlists and instructions, and 1-shot with netlists and instructions—on the **subset of the dataset with associated netlists (Questions With Netlists – 79 templates)**. Additionally, it includes results for the 1-shot prompt on the **subset without netlists (Questions W/O Netlists – 23 templates)**. This breakdown helps identify which types of errors are most prevalent across different prompt configurations and for questions with associated netlists versus questions without netlists.

| Dataset Subset | Prompt | GPT 4o Response Error Rate (%) by Error Type | | | | |
|---|---|---|---|---|---|---|
| | | Math | Formatting | Reasoning | Topology | Direction |
| Questions With Netlists | 0-s w/ netlists | 7.1 | 1.3 | 58.5 | 36.2 | 4.1 |
| | 1-s w/ netlists | 8.4 | 0.5 | 61.8 | 39.2 | 3.5 |
| Questions W/O Netlists | 1-s | 1.7 | 0.0 | 34.8 | 16.5 | 4.4 |

Table 5: **Category Accuracies from Human Analysis of GPT-4o Responses.** The table shows the global accuracy on subsets of GPT-4o responses across four categories (Analog, Basic, Power, and Radio Frequency), based on human analysis results. It presents results on two subsets of the dataset and different prompts, similar to Table 4. The highest accuracy for questions with netlists is bolded.

| Dataset Subset | Prompt | GPT 4o Global Accuracy (%) per Category | | | |
|---|---|---|---|---|---|
| | | Analog | Basic | Power | Radio Frequency |
| Questions With Netlists | 0-s w/ netlists | **30.6** | **49.4** | **30.0** | **20.0** |
| | 1-s w/ netlists | 28.2 | 45.0 | 26.7 | 10.0 |
| Questions W/O Netlists | 1-s | 33.3 | 80.0 | 100.0 | 60.0 |

Table 6: **Level Accuracies from Human Analysis of GPT-4o Responses.** The table shows the global accuracy on subsets of GPT-4o responses across three levels (1, 3, 5), based on human analysis results. It presents results on two subsets of the dataset and different prompts, similar to Table 4. The highest accuracy for questions with netlists is bolded.

| Dataset Subset | Prompt | GPT 4o Global Accuracy (%) per Level | | |
|---|---|---|---|---|
| | | 1 | 3 | 5 |
| Questions With Netlists | 0-s w/ netlists | **49.4** | **31.2** | **18.5** |
| | 1-s w/ netlists | 45.0 | 28.2 | 9.2 |
| Questions W/O Netlists | 1-s | 85.0 | 60.0 | 48.0 |

## AUTHOR CONTRIBUTIONS

Author 1 was responsible for the overall design and execution of the study, including the development of the unique dataset structure, the creation of custom metrics, automatic and human analysis frameworks, and prompt and experiment design. Author 1 also curated the dataset. Author 1 and Author 2 conducted the human analysis of the model responses. Finally, Author 2 and Author 3 reviewed the dataset to ensure its quality and consistency. Authors 4, 5, and 6 provided oversight

and guidance throughout the study, with Author 6 serving as the principal investigator and Authors 4 and 5 offering key support in a supervisory capacity from the industry side.

## ETHICAL CONSIDERATIONS

We addresses the critical points related to ethical considerations, ensuring that our research is conducted responsibly and transparently.

**Data Collection and Privacy:** Our dataset did not involve personal data, ensuring no privacy concerns; however, the dataset will not be shared publicly until informed consent from the authors of sources listed in Appendix A is obtained.

**Use of LLMs for Writing Assistance:** Chat GPT was used to refine the clarity and conciseness of our paper.

## ACKNOWLEDGEMENTS

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

## A DATASET SOURCES

Problem statements and diagrams from the following sources were selected and modified to allow for multiple numerical setups:

1. Gray, P. E., Hurst, P. J., Lewis, S. H., & Meyer, R. G. *Analysis and Design of Analog Integrated Circuits*.

2. Massachusetts Institute of Technology. MIT OpenCourseWare: 6.01 Introduction to Electrical Engineering and Computer Science I.

3. Massachusetts Institute of Technology. 6.002 Circuits and Electronics, Fall 2021.

4. Razavi, B. *RF Microelectronics* (2nd ed.). Prentice Hall.

5. Author 1 *Original problems*.

6. Massachusetts Institute of Technology. MIT OpenCourseWare: 6.061/6.690 Introduction to Power Systems.

7. Analog Devices. *Real Analog - Circuits 1-12*.

8. Bowick, C. *RF Circuit Design* (2nd ed.).

9. Kassakian, J. G. *Principles of Power Electronics* (2nd ed.).

10. Razavi, B. *Design of Analog CMOS Integrated Circuits*.

## B NETLISTS

If a model is given a netlist in the prompt, we give it NETLIST_INSTRUCTIONS_START to which we concatenate only the relevant explanations from the NETLIST_INSTRUCTION_DICTIONARY. The traditional netlist syntax does not accommodate in simplicity for certain elements in our circuit diagrams, hence we adapt it as shown below. Some netlists contain comments, so we concatenate their explanations as well, as necessary.

```
NETLIST_INSTRUCTION_START = "To better understand the given circuit diagram also take a look at the following \
    netlist-like description of the circuit. \
    Some elements and nodes are have no label/name on the diagram, but they are given names in \
    the netlists. Note that <value> of an element in the netlist is given in standard units and it is optional \
    (it can be included but does not have to be in the netlist descritption of an element).\n\
    In the netlist, the elements are listed as:\n"

NETLIST_INSTRUCTION_DICTIONARY = {
    "R"    : "Resistor: R<string> node_1 node_2 <value>\n",
    "C"    : "Capacitor: C<string> node_1 node_2 <value>\n",
    "L"    : "Inductor: L<string> node_1 node_2 <value>\n",
    "V"    : "Voltage source: V<string> node_+ node_- <value>\n",
    "I"    : "Current source: I<string> node_from node_to <value>\n",
    "S"    : "Simple switch: S<string> node_1 node_2\n",
    "D"    : "Diode: D<string> n_anode n_cathode\n",
    "H"    : "Current-controlled voltage source: H<string> node_+ node_-
                <name of the current controlling the source> <transresistance>\n",
    "G"    : "Voltage-controlled current source: G<string> node_1 node_2
                <name of the voltage controlling the source> <transconductance>\n",
    "OPA"  : "Simple Op-Amp: OPA<string> node_output node_input_+ node_input_- <gain (optional)>\n",
    "M"    : "MOS Transistor: M<string> n_drain n_gate n_source n_body NMOS/PMOS\n",
    "Q"    : "BJT: Q<string> n_collector n_base n_emitter PNP/NPN\n",
    "Y"    : "Anonymous element: Y<string> node_1 node_2\n",
    "K"    : "Mutual Inductors: K<string> <inductor1> <inductor2>
                <number of turns in inductor1>:<number of turns in inductor2>\n"
    }

NETLIST_INSTRUCTION_INLINE_COMMENT = "The netlist contains inline comments labeled with \";\", \
    mostly indicating voltages or currents labeled on the diagram. If there is a minus sign, that means \
    the voltage or the current is measured in the opposite direction from the nodes listed for that \
    element. For example, if there is a line \"E N1 N2 ; v\", the voltage v is measured node N1 to N2. \
    That is, N1 is the positive node of the measured voltage v, and N2 is the negative node. \
    On the other hand, if there is a line  \"E N1 N2 ; -v\", the voltage v is measured node N2 to N1. \
    If the comment is about a current, it is the current measured through the listed element E flowing from \
    N1 to N2 if there is a line \"E N1 N2 ; i\", and flowing from N2 to N1 if there is a line \"E N1 N2 ; -i\"\
    Furthermore, note that the positive current direction is into drain node for an NMOS element and \
    out of drain node for a PMOS element.\n"

NETLIST_INSTRUCTION_COMMENT = "The netlist also includes comments marked with \"*\".\n"
```

## C PROMPT DESIGN

Here we explain the prompt design.

System instructions begin with the following 0-shot prompt which describes the desired final answer format:

```
You are an electrical engineering expert.  Solve a given problem step by
step.  At the end of your solution, write "Final Numerical Answer:  N"
where N is your final numerical answer.  If the problem did not have
enough information needed to solve it, put "Unknown" in place of N.
If the problem setup is invalid, and thus the problem does not have
a solution, put "None" in place of N. The final numerical answer, if
different from Unknown and None, should be with precision up to 6 decimal
places.  The numerical answer should be a decimal number with 6 digits
after the decimal point.  Don't write fractions or numbers in any other
format.  Don't write any further explanations after the Final Numerical
Answer.
Here is an example of the answer format:
Question:
What is x = 2 + 2 * 2?
Step by step solution:
Following the PEMDAS rule, we first multiply 2 * 2 = 4.  Then, we add 2 +
4 to get x = 6.
Final Numerical Answer:  6.000000
```

To create a 1-shot prompt from a 0-shot prompt, we add an example problem that is similar to the CIRCUIT data. Depending on whether the model in the experiment was given netlists or not, the model would receive one of the following versions of the problem's solution:

```
# Version 1:  No netlist given in the prompt

Here is an example problem and solution:
Example Problem:
Consider the circuit in the example diagram.  Determine v in Volts.

Solution:
We are asked to find the voltage v across the current source in the
figure.  We can see in the figure that the circuit consists of a
current source I₁ and a resistor network.  If we can find the equivalent
resistance of the resistor network, we can determine the voltage v using
Ohm's law.  From the figure, we can see that R₁ and R₂ are connected
in parallel.  Their combination is connected in series to a parallel
combination of R₃ and R₄.  And this parallel combination is connected in
series with R₅.
```

Therefore, we find that $R_{eq} = R_{12} + R_{34} + R_5 = R_{12} + R_{34} + 100\Omega$. Since $R_1$ and $R_2$ are parallel to each other, we find that $R_{12} = \frac{1}{\frac{1}{R_1} + \frac{1}{R_2}} = \frac{1}{\frac{1}{500} + \frac{1}{500}} = \frac{1}{\frac{2}{500}} = \frac{500}{2} = 250\Omega$. Similarly, $R_{34} = \frac{1}{\frac{1}{R_3} + \frac{1}{R_4}} = \frac{1}{\frac{1}{300} + \frac{1}{100}} = \frac{1}{\frac{100+300}{100(300)}} = \frac{30000}{400} = 75\Omega$.

Thus, $R_{eq} = R_{12} + R_{34} + R_5 = 250\Omega + 75\Omega + 100\Omega = 425\Omega$.

Using Ohm's Law, we find that $v = I_1 R_{eq} = 2A(425\Omega) = 850V$.

```
Final Numerical Answer:  850.000000

# Version 2:  Netlist given in the prompt

Here is an example problem and solution:
Example Problem:
Consider the circuit in the example diagram.  Determine v in Volts.
Netlist:
```

```
```
I1 0 N1 2
R1 N1 N2 500
R2 N1 N2 500
R3 N2 N3 300
R4 N2 N3 100
R5 N3 0 100
```
```

Solution:
We are asked to find the voltage $v$ across the current source in the
figure. We can see in the figure that the circuit consists of a
current source $I_1$ and a resistor network. If we can find the equivalent
resistance of the resistor network, we can determine the voltage $v$ using
Ohm's law. From the figure, we can see that $R_1$ and $R_2$ are connected
in parallel. Their combination is connected in series to a parallel
combination of $R_3$ and $R_4$. And this parallel combination is connected
in series with $R_5$. We confirm this in the netlist. R1 and R2 share two
same nodes N1 and N2, so they are connected in parallel. R3 and R4 share
two same nodes N2 and N3, so they are connected in parallel. R1, R2,
R3, and R4 share node N2, so the parallel combinations R12 and R34 are
connected in series. Finally, R3, R4, and R5 share a node N3, so the
parallel combination R34 and R5 are connected in series. Therefore, we
find that $R_{eq} = R_{12} + R_{34} + R_5 = R_{12} + R_{34} + 100\Omega$. Since $R_1$ and $R_2$ are parallel
to each other, we find that $R_{12} = \frac{1}{\frac{1}{R_1} + \frac{1}{R_2}} = \frac{1}{\frac{1}{500} + \frac{1}{500}} = \frac{1}{\frac{2}{500}} = \frac{500}{2} = 250\Omega$.
Similarly, $R_{34} = \frac{1}{\frac{1}{R_3} + \frac{1}{R_4}} = \frac{1}{\frac{1}{300} + \frac{1}{100}} = \frac{1}{\frac{100+300}{100(300)}} = \frac{30000}{400} = 75\Omega$. Thus,
$R_{eq} = R_{12} + R_{34} + R_5 = 250\Omega + 75\Omega + 100\Omega = 425\Omega$. Using Ohm's Law, we find
that $v = I_1 R_{eq} = 2A(425\Omega) = 850V$.

Final Numerical Answer: 850.000000

The model would also be given the 1-shot example diagram.

The models would then receive the problem template and numerical setup. For example, if we were
asking the model to solve the problem in Figure 2, we would add:

Calculate the current $I$ in Amperes in the given circuit.
Assume $V = 5V$ and $R = 100\Omega$.

If the model was provided with a netlist, it would additionally receive the necessary netlist explana-
tions detailed in Appendix B as well as the netlist. For the above example, that would be:

To better understand the given circuit...
(the rest of netlist instructions)
The netlist:
V N1 0
R N1 0 ; I

## D  ERROR TYPES

We categorized errors made by the model into the following types:

- **Math Errors**: Any type of mistake related to mathematical computation, precision, or misunderstanding of mathematical concepts. This includes errors in basic arithmetic, formula application, or incorrect assumptions about numerical values.

- **Formatting Errors**: These occur when the model outputs an answer in an incorrect format (correct format detailed in Appendix C). For example, the model may respond with "FNA: ..." instead of using the correct label, "Final Numerical Answer:" or might misapply other required conventions.

- **Reasoning Errors**: Any mistake that is not a Math or a Formatting error. These include two subcategories:

  - **Topology Errors**: The model misunderstood the structural layout or connection of elements in the circuit. For instance, it could misidentify an element or incorrectly interpret how components are interconnected within the system.
  - **Direction Errors**: The model misunderstood the orientation of an element, the direction of a current, etc. This involves confusion over how elements are aligned or how power/current flows through the circuit.

The main error types are Math, Formatting, and Reasoning. That means, all direction and topology errors count as reasoning errors.

864
865
866
867
868
869

# E  QUALITATIVE ANALYSIS EXAMPLES

870
871
872
873
874
875
876
877
878
879
880
881
882
883
884
885
886
887
888
889

Here, we present several concrete qualitative analysis examples. Portions of GPT's responses have been redacted for clarity or brevity, indicated by ellipses ("...").

## E.1  MISUNDERSTANDING TOPOLOGIES

GPT had trouble understanding even slightly complex topologies. Sometimes, netlists would not help its understanding.

Consider the problem ID 46 adapted from MIT 6.002 (source listed in Appendix A):

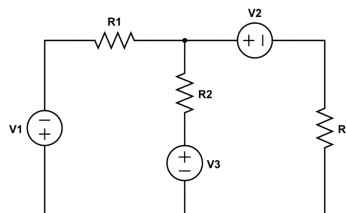

```
Template:
Find the power in Watts delivered
to R_3 in the circuit in the figure.

Numerical Setup:
Assume that V_1 = 80 V, V_2 = 100 V,
V_3 = 40 V, R_1 = 2 Ohm, R_2 = 12 Ohm,
and R_3 = 4 Ohm.
```

890
891
892
893
894
895

In experiment 5, GPT was given the 0-shot prompt, and we can see the topology misunderstanding from the KVL equations in the response below. The left loop does not include $R_3$ but does include another resistor and another voltage source. Additionally, the signs in the KVL equations are inconsistent.

896
897
898
899
900
901
902
903
904
905
906
907
908
909
910
911
912
913
914
915

---

EXPERIMENT 5, QUESTION ID 46

GPT's RESPONSE:

Let's analyze and solve for the power delivered to $R_3$ in
the given circuit step by step.
Step 1: Identify the Loop Equations

We will use Kirchhoff's Voltage Law (KVL) for the two
loops with given voltage sources and resistors.
Loop 1 (left loop):

$$-V_1 + I_1 R_1 + I_3 R_3 = 0$$

Loop 2 (right loop):

$$V_2 + I_2 R_2 - V_3 - I_3 R_3 = 0$$

Step 2: Write the Equations in Terms of Currents
Expressing the above in terms:
...

---

916
917

A similar misunderstanding can be seen in experiment 7 where GPT was given the 1-shot prompt. The left-loop is missing an element and the signs in the KVL equations are inconsistent.

```
EXPERIMENT 7, QUESTION ID 46

GPT's RESPONSE:

First, we need to find the currents through the resistors
in the given circuit.
1. Apply Kirchhoff's Voltage Law (KVL) around the left
loop (containing V₁, R₁, and R₂):
```

$$V_1 - I_1 R_1 - V_3 = 0$$

```
...
2. Apply Kirchhoff's Voltage Law (KVL) around the right
loop (containing V₃, R₂, V₂, and R₃):
```

$$V_3 + I_1 R_2 - V_2 - I_2 R_3 = 0$$

```
...
```

Looking at experiment 6, where GPT was given the 0-shot prompt and the netlist, GPT could not reason about this topology even with a netlist. In step 2, it aready makes some mistakes: $V_1 = -80V$, $V_3 \neq 0$, $V_4 \neq -40V$, and $N_4$ is not directly connected to $V_3$.

```
                    Netlist given to the model:
                         V1 0  N1
                         R1 N1 N2
                         R2 N2 N3
                         V3 N3 0
                         V2 N2 N4
                         R3 N4 0
```

```
EXPERIMENT 6, QUESTION ID 46

GPT's RESPONSE:

1. Nodal Analysis:
Let's define the following nodes:
- V₀ = 0 V
- V₁ is the potential of node N1
- V₂ is the potential of node N2
- V₃ is the potential of node N3
- V₄ is the potential of node N4

From the netlist and the given values:
- V1 = 80 V
- V2 = 100 V
- V3 = 40 V
- R1 = 2 Ω
- R2 = 12 Ω
- R3 = 4 Ω

2. Node Voltages:
- V₁ is at 80 V because it's directly connected to V1.
- V₃ = 0 V because it's connected to the ground.
- V₄ is at -40 V because it's directly connected to V3.
...
```

Sometimes, when netlists were provided, GPT would not explicitly use them.

Consider the topology in problem IDs 121-125, focusing on the setup in ID 121. The topology consists of a voltage source connected to a resistor network, and the model is asked to find the total

current supplied by the voltage source. GPT had a difficult time understanding the complex resistor network with and without netlists. Interestingly, its responses do not indicate the use of netlists.

In experiment 5, GPT was given the 0-shot prompt and could not understand the resistor connections, which can be seen from the response below (displayed without equations for easier following).

```
EXPERIMENT 5, QUESTION ID 121

GPT's RESPONSE:

Step by step solution:
1. Calculate the equivalent resistance of R₂ and R₃ in
parallel:
...
2. Add R₄ in series with R₂₃:
...
3. Calculate the equivalent resistance of R₅ and R₆ in
parallel:
...
4. Calculate the equivalent resistance of R₇ and R₈ in
parallel:
...
5. Add R₇₈ in series with R₆:
...
6. Combine R₂₃₄ and R₆₇₈ in parallel:
...
7. Add R₁ in series with the equivalent resistance R₂₃₄₆₇₈:
...
8. Use Ohm's Law to find the current I: ...
Final Numerical Answer: 4.342778
```

In experiment 6, GPT was given the 0-shot prompt and the netlist. It misunderstood the resistor network, and it did not explicitly use the provided netlist (displayed without equations for easier following).

```
EXPERIMENT 6, QUESTION ID 121

GPT's RESPONSE:

1. Combine R₂ and R₄ (parallel combination):
...
2. Combine R₂₄ and R₃ (series combination):
...
3. Combine R₂₄₃ and R₅ (series combination):
...
4. Combine R₇ and R₈ (parallel combination):
...
5. Combine R₇₈ and R₆ (series combination):
...
6. Combine R₂₄₃₅ and R₇₈₆ (parallel combination):
...
7. Combine R₁ and R_{total_branch} (series combination):
...
8. Use Ohm's Law to find the current I:
...

Final Numerical Answer: 4.399751
```

Similar can be observed in experiments 7 and 8, where GPT was given the 1-shot prompt without and with the netlist respectively; however, it did not explicitly use the netlist given in experiment 8.

## E.2 Utilizing netlists successfully

Netlists sometimes helped GPT successfully reason about a topology.

Consider the problem ID 174. The topology consists of a current source, a voltage source, and two resistors, and the model is asked to find the current through one of the resistors. In experiment 5, GPT-4o was given the 0-shot prompt and misunderstood the topology of this circuit, particularly confusing current directions. In experiment 7, GPT-4o was given the 1-shot prompt and misunderstood the topology of this circuit, interpreting resistors to be connected in series. However, in experiment 6, GPT-4o was given the 0-shot prompt and the netlist for this circuit, which it utilized to successfully understand the topology, as shown below in GPT's response.

---

EXPERIMENT 6, QUESTION ID 174

GPT's RESPONSE:

```
Step by step solution:
```

1. From the netlist:
- A current source $U = 0.6\,A$ is connected between ground (0) and node $N1$.
- $R1 = 10\,\Omega$ is connected between $N1$ and ground (0).
- $R2 = 20\,\Omega$ is connected between $N1$ and $N2$.
- A voltage source $V = 30\,V$ is connected between ground (0) and $N2$.

2. Current $U = 0.6\,A$ flows from ground to $N1$.

3. Apply Kirchhoff's Current Law (KCL) at node $N1$:

$$I_U = I_{R1} + I_{R2}$$

where
- $I_U$ is the current from the current source, which is $0.6\,A$.
- $I_{R1}$ is the current through $R1$.
- $I_{R2}$ is the current through $R2$.

4. Calculate the current through $R1$:

$$I_{R1} = \frac{V_{N1}}{R1} = \frac{V_{N1}}{10}$$

5. Calculate the current through $R2$:

$$I_{R2} = \frac{V_{N1} - V_{N2}}{R2} = \frac{V_{N1} - 30}{20}$$

Using KCL:

$$0.6 = \frac{V_{N1}}{10} + \frac{V_{N1} - 30}{20}$$

6. Simplify the equation to solve for $V_{N1}$:
Multiply through by 20 to clear the denominators:

$$12 = 2V_{N1} + V_{N1} - 30$$

$$12 = 3V_{N1} - 30$$

Add 30 to both sides:

$$42 = 3V_{N1}$$

Divide by 3:

$$V_{N1} = 14\,V$$

7. Calculate $I_{R1}$, the current through $R1$:

$$I_{R1} = \frac{V_{N1}}{10} = \frac{14}{10} = 1.4\,A$$

8. The current $I$ labeled in the diagram is the current through $R1$, which is $I_{R1}$.

```
Final Numerical Answer: 1.400000
```

---

### E.3 MISUNDERSTANDING DIRECTIONS

GPT would sometimes misunderstand element orientations, such as the positive and negative nodes of a voltage source, or current directions.

Consider the problem ID 179. In experiment 8, GPT was given a 1-shot prompt and a netlist. However, it misunderstood the current source orientation – the current is flowing from ground to N2.

---

EXPERIMENT 8, QUESTION ID 179

GPT's RESPONSE:

```
...
1. From the netlist and the circuit diagram (Figure
134-01): - U = 0.4V - R1 = 4Ω - I1 = 1A (current flowing
from N2 to ground through the current source)
...
```

---

Consider the problem ID 31. In experiment 6, GPT was given a 0-shot prompt and a netlist. However, it misunderstood the labeled current directions, as can be seen in its KCL application. The correct equation is $i_3 = i_2 + i_4$.

---

EXPERIMENT 6, QUESTION ID 31

GPT's RESPONSE:

```
...
Step 3: Write the node voltage equation at node N2 using
Kirchhoff's Current Law (KCL):
```

$$i_3 + i_4 = i_2$$

```
...
```

---

In the same experiment, consider the problem ID 47. GPT misunderstood voltage source $V_1$ orientation and wrongly deduced a node potential (further topology misunderstandings can be seen as well).

---

EXPERIMENT 6, QUESTION ID 47

GPT's RESPONSE:

```
To find the power delivered to R₃, we need to determine
the voltage across it and the current through it.

1. Nodal Analysis:
Let's define the following nodes:
- V₀ = 0 V
- V₁ is the potential at node where V₁ is connected.
- V₂ is the potential at the node where V₂, R₁ and R₂ are
connected.

2. Assign Potential Values:
- Node at ground: V₀ = 0 V
- Node N1: V(N1) = 20 V from V₁
- Node N2 to ground V(N2,0) = 40 V from V₃
...
```

---

## E.4 MINOR REASONING ERRORS

GPT would sometimes correctly solve a problem but display minor reasoning errors in the process, resulting in human evaluation labeling its response as incorrect.

Consider the problem ID 15. The topology consists of a voltage source and a small resistor network, and the model is asked to find the current through one of the resistors. In experiment 8, GPT was given a 1-shot prompt and a netlist. Its solution was overall correct. However, it wrongly claimed that the voltage drop across the parallel combination of $R_2$ and $R_3$ is the same as the voltage across $R_1$. This could be interpreted as a conceptual misunderstanding of voltage dividers. It didn't use this wrong conclusion, so its final answer was correct.

---

EXPERIMENT 8, QUESTION ID 15

GPT's RESPONSE:

```
To determine the current i₃ in the given circuit, we can use various circuit analysis
techniques. Let's follow Ohm's Law and the principles of series and parallel circuits.
1. Given Data:
```
- $V_1 = 10V$
- ...

```
2. Identify Series and Parallel Connections:
```
- Voltage source $V_1$ is connected to $R_1$ and the combination of $R_2$ and $R_3$.

```
3. Calculating Equivalent Resistance:
Since R₂ and R₃ are connected in parallel, we calculate their equivalent resistance
(R₂₃):
```

$$\frac{1}{R_{23}} = \frac{1}{R_2} + \frac{1}{R_3}$$

...

```
4. Total Resistance in the Circuit:
Now, calculate the total resistance in the circuit, which is the sum of R₁ and R₂₃:
```

$$R_{total} = R_1 + R_{23}$$

...

```
5. Total Current i_total:
Using Ohm's Law, the total current i_total supplied by the voltage source V₁ is:
```

$$i_{total} = \frac{V_1}{R_{total}}$$

...

```
6. Current through R₂ and R₃ (Parallel Combination):
The voltage drop across the parallel combination of R₂ and R₃ is the same as the
voltage across R₁.
Voltage drop across R₁:
```

$$V_{R1} = i_{total} \times R_1$$

$$V_{R1} = 0.750000 \times 5 = 3.750000V$$

```
Hence, voltage across R₂ (which is same as V_R3):
```

$$V_{R2} = V_{total} - V_{R1}$$

$$V_{R2} = 10 - 3.750000 = 6.250000V$$

```
7. Current i₃ through R₂:
Using Ohm's law again for R₂:
```

$$i_3 = \frac{V_{R2}}{R_2}$$

$$i_3 = \frac{6.250000}{10}$$

$$i_3 = 0.625000A$$

```
Final Numerical Answer: 0.625000
```

## E.5 MATH ERRORS

Most math errors GPT made were approximation errors, usually in division and exponent and logarithm calculations. Occasionally, GPT also made small errors in equation manipulation and computation.

Consider the problem ID 444. The model was asked to find the small-signal output resistance of a given circuit. In experiment 8, GPT was given a 1-shot prompt and a netlist. GPT makes a computation mistake in the last calculation step concluding $R_{out} = \frac{1}{250 \times 10^{-3} \text{ S}} = \frac{1}{0.25}$ k $= 4$ k.

---

EXPERIMENT 8, QUESTION ID 444

GPT's RESPONSE:

```
...
```

$$R_{out} = \frac{1}{250 \times 10^{-3} \text{ S}} = \frac{1}{0.25} \text{ k} = 4 \text{ k}$$

```
Therefore, the small-signal output resistance $R_{out}$ is:

Final Numerical Answer: 4.000000
```

---

Consider the problem ID 265. The model was asked to find the linear scale loss in a given circuit. In experiment 6, GPT was given a 0-shot prompt and a netlist. In the last step, it made an approximation, which resulted in the final answer being outside the allowed error range.

---

EXPERIMENT 6, QUESTION ID 265

GPT's RESPONSE:

```
...
```

$$= \frac{6806.25}{6187.5} \approx 1.099$$

```
Final Numerical Answer: 1.099000
```

---

Consider the problem ID 326. The model was asked to determine the overall noise figure of a given circuit. In experiment 8, GPT was given a 1-shot prompt and a netlist. GPT had a small error in the logarithm calculation.

---

EXPERIMENT 8, QUESTION ID 326

GPT's RESPONSE:

```
... 4. Convert Back to Noise Figure in dB:
```

$$NF_{\text{total}} = 10 \log_{10}(1.904893) = 10 \times 0.280173$$

$$NF_{\text{total}} \approx 2.801732 \text{ dB}$$

```
Final Numerical Answer: 2.801732
```

---

## E.6 NOT FOLLOWING INSTRUCTIONS, NOT ADAPTING TO THE CONTEXT

Sometimes, GPT would not follow the given instructions or fail to consider the problem context.

Consider the problem ID 71. In experiment 6, GPT was given the 0-shot prompt and the netlist. The topology consists of a capacitor connected to a simple resistor network, and the model is asked to find the voltage across the capacitor. GPT claims it was not given the value of $t$, even though the value was given in the setup. GPT also failed to follow the formatting instructions in the prompt – the model should format the answer as `Final Numerical Answer:Unknown` in the case that insufficient information was provided.

EXPERIMENT 6, QUESTION ID 71

GPT's RESPONSE:

```
...
Since the solution requested is to find $v_b(t)$ and an
explicit time was not provided, the final answer must be
left in the general solution form, which is:
```
$v_b(t) = 5 \exp(-2t)$

```
Therefore, for a general time $t$:

Final Numerical Answer:
```

$v_b(t) = 5 \exp(-2t)$
```
...
```

Consider the problem ID 486. In experiment 8, GPT was given the 1-shot prompt and the netlist. The model fails to recognize the correct sign of the gain. Instead, it applied the standard common-source amplifier gain expression without adapting it to the specific circuit context.

EXPERIMENT 8, QUESTION ID 486

GPT's RESPONSE:

```
To determine the small signal voltage gain of the circuit in figure 242-01, we need to
analyze the small signal equivalent circuit. Given:
- ...
- Assuming that $M_1$, $M_2$, and $M_3$ have the same width and length.

We'll follow these steps to calculate the voltage gain:
1. Identify the operating principles and small-signal parameters.
2. Construct the small-signal model.
3. Calculate the small-signal gain.

Step 1: Operating Principles and Small-Signal Parameters
- $M_1$ is an NMOS transistor.
- $M_2$ and $M_3$ are PMOS transistors used in current mirror configuration.
- $g_m$ represents the transconductance of the transistor $M_1$.

Step 2: Small-Signal Model

The small-signal equivalents can be represented as follows:
- $M_1$ operates with a transconductance $g_{m1} = 10\,mS$.
- $M_2$ and $M_3$ form a current mirror.

Step 3: Gain Calculation
1. Small-Signal Equivalent Circuit:
- ...

2. Small-Signal Resistance at Drain of $M_3$:
...

3. Small-Signal Current through $M_1$:
...

4. Current Mirror (PMOS):
The current through $M_2$ is mirrored by $M_3$.

5. Small-Signal Voltage Gain:
The voltage gain $A_v = \frac{V_{out}}{V_{in}}$ can be computed as:
```

$$A_v = -g_{m1} \cdot R_L$$

```
Given:
```
$$g_{m1} = 10\,mS = 0.01\,S$$
```
and
```
$$R_L = 100\,\Omega$$
```
we have:
```
$$A_v = -(0.01\,S) \cdot (100\,\Omega)$$
```
Simplifies to:
```
$$A_v = -1$$

```
Final Numerical Answer:
```
$$Final Numerical Answer : -1.000000$$

E.7 HALLUCINATIONS

Sometimes, GPT would introduce nonsensical conclusions throughout its solution. These conclusions usually seem like hallucinations about common configurations.

Consider the problem ID 199. The topology features an op-amp in a negative feedback configuration, and the model is ask to find the current through one of the resistors in the circuit. In experiment 6, GPT was given the 0-shot prompt and the netlist. The model eventually produces the correct solution. However, in its solution, it says that the "voltage drop across the $100\,\Omega$ resistor is not needed since $V_p$ can be assumed to be at virtual ground if $V_n = V_p$." This is a common configuration for op-amps in negative feedback, but it is not true for the given topology, and the model does not use it later in the solution.

---

EXPERIMENT 6, QUESTION ID 199

GPT's RESPONSE:

```
To calculate the current through resistor R₁ in the given
circuit , follow these steps:

Step-by-Step Solution:

1. Identify Voltages and Nodes:
- The non-inverting input of the op-amp (Vₚ) is connected
to node N1 through a 100  resistor and a 10 V source.
- The inverting input of the op-amp (Vₙ) is connected to
node Vₙ.
- Node N3 is the output node of the op-amp.

2. Op-Amp Ideal Assumptions: - For the ideal op-amp,
the voltage at the inverting input (Vₙ) is equal to the
voltage at the non-inverting input (Vₚ).
- The input differential voltage is zero, so Vₙ = Vₚ.

3. Calculate Vₚ: - Since the non-inverting input (Vₚ) is
connected to the junction of the 100  resistor and the 10
V source, we need to find this voltage.
- Voltage drop across the 100  resistor is not needed
since Vₚ can be assumed to be at virtual ground if Vₙ = Vₚ.

4. Voltage at Vₚ:
- Given Vₚ directly after the resistor is simply 10 V
considering ideal conditions of the circuit.

5. Calculate Vₙ:
- Given Vₙ = Vₚ = 10 V

6. Use Ohm's Law to determine current through R₁:
- Vₙ is at 10 V, and one side of resistor R₁ is at 0 V
(GND).
- Current through R₁ is calculated by I_R1 = Vₙ/R1.

Calculation:
```
$$I_{R1} = \frac{V_n}{R1} = \frac{10\,V}{2\,\Omega} = 5\,A$$

```
Final Numerical Answer:
Final Numerical Answer: 5.000000
```

---

In the same experiment and concerning the same topology in problem ID 197, GPT hallucinated that $V_p$ is determined by a voltage divider. This is also a common configuration for op-amps in negative feedback. Similarly as ID 199, GPT did not use this conclusion in the solution, so its final answer was correct.

