# OpenReview forum: "CIRCUIT: A Benchmark for Circuit Interpretation and Reasoning Capabilities of LLMs"
_ICLR.cc/2025/Conference — Submitted to ICLR 2025_

### Official Review · Reviewer_Ahcw · 2024-11-02

**Soundness:** 2
**Presentation:** 2
**Contribution:** 2
**Rating:** 5
**Confidence:** 5

**Summary:**

This paper introduces a benchmark designed to evaluate the circuit interpretation and reasoning capabilities of large language models (LLMs).
The benchmark comprises 510 question-answer pairs covering various areas of analog circuit design.
Evaluations conducted on three representative LLMs provide some insights into the reasoning abilities of state-of-the-art models.

**Strengths:**

The paper presents an intriguing study on the capabilities of large language models (LLMs) in circuit interpretation and reasoning and offers some insights to advance research in leveraging LLMs for analog design automation.

The dataset, comprising over 500 question-answer pairs sourced from a variety of courses and textbooks, appears substantial and non-trivial, providing a solid foundation for evaluating LLM performance in this domain.

**Weaknesses:**

The paper’s technical contributions appear limited, focusing primarily on prompt engineering.

Additionally, the benchmark used raises concerns: although it consists of 500 question-answer pairs, these questions are sourced from textbooks, which overly simplify the real-world challenges of analog circuit design. Analog circuit design involves multiple critical stages, such as topology generation, device sizing, and layout design, and it is unclear how the current benchmark addresses these complexities.

Furthermore, the results seem unreasonable. Instead of reporting only the overall accuracy, a more detailed breakdown of circuit performance across varying complexities should be presented. Understanding how the model handles complex analog circuit problems, as opposed to simpler ones, would provide far more meaningful insights into its effectiveness.

The insights are limited. It does not show how to improve the capabilities of LLM in addressing analog design automation.

**Questions:**

How does the benchmark address the different design stages of analog circuits?
What is the success ratio of each level circuit (e.g., 1, 3, 5) based on the benchmark?

---

> ### Author Response · Authors · 2024-11-20
> **Clarifications on Dataset Scope and Design, Technical Contributions, and Model Performance Analysis**
>
> Thank you for your thoughtful comments and questions. We address your points as follows:
>
> **Questions:**
> 1. **Different stages of analog circuit design:**
> While our dataset does not address higher-level design tasks, it establishes a foundational framework for evaluating topology understanding. Our benchmark reveals that LLMs struggle significantly with understanding basic topologies, which we consider a precursor to performing any complex design task. Without this foundational understanding, LLMs are unlikely to perform reliably on higher-level tasks. In future work, we aim to first address these fundamental challenges and then expand the dataset with more advanced analog design tasks. The structure of our dataset enables straightforward expansions for scalable and reliable automatic testing of LLMs’ capabilities in more advanced design stages. We view our work as a necessary first step, focusing on understanding topologies, which underpins all subsequent design processes. We will work on emphasizing these points in the revision.
> 2. **Per-level accuracies:**
> We acknowledge the importance of analyzing performance across levels of complexity and provide the following breakdown of GPT-4o's per-level accuracies. These can be included in the revision:
>
> | Dataset Subset | Prompt            |     |  GPT 4o Accuracy (%) per level       |       |
> |------------------------|-------------------|:-------------:|:-------------:|:-------------:|
> |                        |                  |       1       |       3       |       5       |
> | Questions With Netlists| 0-s w/ netlists  |   **49.4**    |     31.2      |     18.5      |
> |                        | 1-s w/ netlists  |     45.0      |     28.2      |      9.2      |
> | Questions W/O Netlists | 1-s              |     85.0      |     60.0      |     48.0      |
>
> Further insights can be drawn from Appendix E which contains an analogous table of per-category accuracies. We will also include a discussion of these results in the revision.
>
> **Weaknesses:**
> 1. **Technical contributions:**
> While prompt engineering was an important aspect of our work, a key contribution lies in the dataset design itself, which enables an automated, unit-test-like evaluation of reasoning tasks. By abstracting circuit reasoning into structured templates, we ensured a reliable and scalable evaluation process. Additionally, our pass@k/n metric provides nuanced insights into model capabilities, demonstrating how reasoning tasks can be reliably evaluated, without human or LLM involvement, offering deeper insights. These aspects set a foundation for future work on scalable, reliable, and comprehensive automatic evaluation of reasoning tasks. We will make sure to emphasize these points in the revision.
> 2. **Dataset scope:**
> We agree that analog circuit design involves more complex stages than what is addressed in our benchmark. However, our dataset was intentionally designed as a starting point, focusing on topology understanding. Initial results reveal that LLMs struggle significantly with understanding basic topologies, which we consider a precursor to performing any complex design task. In future work, we aim to first address these fundamental challenges and then expand the dataset with more advanced analog design tasks. Our revision will include a clarification of our design choices and how the dataset can be extended to more complex tasks.
> 3. **Reported accuracies:**
> We recognize the value of detailed performance analysis but find that objectively defining complexity is challenging. Currently, we rely on levels and categories, as shown earlier. Qualitative analysis reveals that models perform well only on simpler topologies. Topology complexity can be described using graph-theoretic approaches. However, question complexity may not always align with topology complexity, as the reasoning required to solve a problem can depend on factors beyond the circuit's structural intricacy. Addressing this discrepancy and developing a robust definition of question complexity remains an open challenge for future work.
> 4. **Limited insights:**
> The benchmark results highlight specific limitations in LLM reasoning, which must be addressed before these models can contribute meaningfully to design automation. Our methods for improving LLM capabilities yielded only modest gains. Nevertheless, our analysis revealed more fundamental challenges LLMs face in reasoning tasks. Addressing these foundational issues, for general and circuit-related applications, is crucial before tackling advanced design tasks and shows a promising direction for future work.
>
> We hope this clarifies your questions and addresses your concerns. If you have any further questions or feedback, we would be happy to address them.

---

> > ### Comment · Reviewer_Ahcw · 2024-11-23
> > **Thanks for your response.**
> >
> > Thanks for authors in addressing my questions. Although I think authors did a good job in characterizing the reasoning ability of LLMs in circuit design, isn't this obvious? LLMs mainly focus on text learning, while circuit topologies can be described by many modalities. More importantly, the authors do not show insight on how to improve the reasoning ability of LLMs in circuit topology understanding.

---

> ### Author Response · Authors · 2024-11-23
> **Clarifications on Contributions and Novelty of Our Work**
>
> Thank you for your follow-up and for recognizing our efforts in characterizing the reasoning ability of LLMs in circuit design.
>
> We would like to address your concerns in detail:
>
> **The findings:**
> While it may seem intuitive that LLMs, primarily trained on text, would struggle with tasks involving circuit topologies—a domain that requires multi-modal reasoning—our work provides the first empirical evidence quantifying the extent and nature of these challenges. This was not a foregone conclusion, as state-of-the-art LLMs have demonstrated unexpected capabilities in other reasoning tasks. By systematically identifying specific shortcomings, our study moves beyond intuition to offer actionable insights for targeted improvements.
>
> **Insights into improving LLM reasoning:**
> We recognize the importance of improving reasoning capabilities, and while our focus was on benchmarking, we did attempt strategies for enhancement. Specifically, we incorporated netlists and explored prompt engineering, which led to modest but measurable improvements. These methods were deliberately chosen for their practicality and applicability, ensuring they are easy for circuit designers to implement without requiring labor-intensive annotations or descriptions. These results indicate that progress is possible, but they also reveal underlying limitations in LLM reasoning that require deeper changes—an important direction for future work.
>
> **Key contributions of our work:**
> Beyond characterizing LLM limitations in reasoning about analog circuit topologies, our work introduces a scalable, unit-test-inspired framework for automatically evaluating reasoning tasks. This structure ensures reliability, minimizes/removes human intervention, and provides a foundation for building more complex datasets and benchmarks. Our evaluation framework is the product of significant effort to abstract complex reasoning challenges into a scalable and reliable evaluation structure, laying the groundwork for advancing reasoning evaluation in various domains, including circuit design. We envision this framework inspiring further research into evaluation techniques for reasoning tasks.
>
> We hope this response clarifies the novelty and significance of our contributions. We appreciate your feedback and welcome further discussion.

---

> > ### Comment · Reviewer_Ahcw · 2024-11-29
> > **Thanks.**
> >
> > Thanks for authors' hard work in addressing my concerns. My key point is that these circuit examples from text books do not represent practical analog IC design. Thus, the impact and use of the circuit benchmark is limited. Yet, I raise my rating to encourage authors to improve the work further.

---

> > > ### Author Response · Authors · 2024-11-29
> > > **Acknowledging Feedback and Future Improvements for CIRCUIT**
> > >
> > > Thank you for your thoughtful follow-up and for raising your rating. We sincerely appreciate your recognition of our effort to address your concerns. Your feedback has been invaluable, and we are excited to leverage the scalability of our dataset to further build on this work and enhance its impact.

---

### Official Review · Reviewer_buQo · 2024-11-02

**Soundness:** 2
**Presentation:** 3
**Contribution:** 2
**Rating:** 5
**Confidence:** 4

**Summary:**

This paper focuses on the application of large language models (LLMs) in analog circuit design. Given that analog circuit design highly depends on human expertise and faces a labor shortage problem, the authors aim to evaluate the capabilities of LLMs in this field by creating the CIRCUIT dataset. The paper details the construction of the dataset, evaluation metrics and methods, and conducts experiments to evaluate multiple LLMs. Finally, it discusses the significance, limitations, and future directions of the research results.

**Strengths:**

1. The article constructs a dataset for analog circuits to evaluate the capabilities of LLM in analog circuits, promoting the exploration and research of LLM in analog circuits.
2. The construction of the dataset is relatively comprehensive, covering different difficulties, and considering diagrams and netlists.
3. The experiment is relatively detailed, testing and validating the current mainstream models and conducting analysis and summary.

**Weaknesses:**

1. The dataset focuses more on evaluating the LLM's answers to analog circuit questions, similar to math problems. However, it cannot verify the LLM's ability to assist in analog circuit design. For example, use a LLM to evaluate the quality of circuit design in terms of power, performance, and area (PPA).
2. The dataset lacks diversity in design and does not cover the entire analog circuit design process. For example, circuit structure design, device selection, test verification and physical design, etc.
3. The benchmark lacks an evaluation of the overall performance of the circuit.

**Questions:**

1. How does the benchmark results help to improve LLM for EDA-aided design?
2. The current dataset focuses more on formula derivation and mathematical calculations, but may lack an understanding of how to evaluate LLM's understanding of the quality of analog circuit design. For example, evaluate the ability of large language models in predicting circuit PPA, ultra-parameter tuning of circuits, and test verification.

---

> ### Author Response · Authors · 2024-11-20
> **Clarifications on Benchmark Scope, Dataset Diversity, and Future Directions**
>
> Thank you for your thoughtful comments and questions. We address your points as follows:
>
> **Questions:**
> 1. **How the benchmark results help LLMs for EDA-aided design:**
> Our results demonstrate that LLMs currently have a very limited understanding of simple circuit topologies. This fundamental limitation must be addressed as a prerequisite to integrating LLMs into the EDA-aided design process. The poor performance observed highlights that LLMs’ usefulness in such workflows would currently be limited and unreliable. Addressing these basic shortcomings in topology understanding is crucial before advancing to more complex design tasks. We shall work on emphasizing these points in our paper in the revision.
> 2. **Dataset scope:**
> While our dataset does not address higher-level design tasks, it establishes a foundational framework for evaluating topology understanding. This structure can be extended to construct datasets for scalable and reliable automatic testing of LLMs’ capabilities in more advanced design stages. We view our work as a necessary first step, focusing on topology comprehension, which underpins all subsequent design processes. Addressing LLMs’ topology understanding limitations and expanding our dataset to more complex design tasks are exciting directions for future work.
>
>
> **Weaknesses:**
> 1. **Dataset scope:**
> We reiterate the points made above. We acknowledge that our dataset does not address the full analog circuit design process. However, it reveals that LLMs struggle significantly with understanding basic topologies, which we consider a precursor to performing any complex design task. Without this foundational understanding, LLMs are unlikely to perform reliably on higher-level tasks. In future work, we aim to first address these fundamental challenges and then expand the dataset with more advanced analog design tasks.
> 2. **Dataset diversity:**
> Our dataset focuses on topology understanding as a starting point and is intentionally limited in scope to address this specific area. We believe the dataset’s structure can be reused to reliably automatically evaluate LLMs on more complex tasks and to expand into areas such as circuit structure design and device selection. These are promising directions for future work.
> 3. **Evaluation of circuit performance:**
> We recognize that the benchmark does not evaluate the overall performance of circuits. This was a deliberate decision to focus on reasoning and topology comprehension, which form the foundation for such evaluations. Expanding the dataset to incorporate performance metrics and end-to-end design tasks is an important future direction, but we believe the fundamental understanding tested in our work must first be established for LLMs to contribute meaningfully to circuit design.
>
> We hope this clarifies your questions and addresses your concerns. If you have any further questions or feedback, we would be happy to address them.

---

### Official Review · Reviewer_5hrA · 2024-11-02

**Soundness:** 3
**Presentation:** 3
**Contribution:** 3
**Rating:** 5
**Confidence:** 3

**Summary:**

This paper presents a Circuit Interpretation and Reasoning Capabilities (CIRCUIT) dataset to evaluate LLMs in understanding and reasoning about analog circuits. The authors conduct a series of experiments to assess the performance of various LLMs in understanding analog circuits and their topologies from diagrams and netlists.

**Strengths:**

1. This paper is the first to evaluate the ability of LLMs in circuit design domain and summarizes the limitations of existing LLMs according to the results. The authors summarize the limitations of existing models based on the experimental results. The qualitative analysis of errors offers valuable insights for improving LLMs.
2. This paper is well organized and clear.

**Weaknesses:**

1. The sources of problems are limited. Most of problems are collected from textbooks published more than 10 years ago, which primarily assess general knowledge of analog circuits. It should be beneficial to construct more challenging problems about complex analog IC designs from various sources, such as handmade questions according to modern IC datasheets.
2. The categories in this dataset are imbalance. In Figure 3, most of Basic questions are in level 1, while RF questions are more challenging. Besides, there is a lack of detailed analysis regarding LLM performance across different levels and categories.

**Questions:**

1. How to differentiate between errors where LLMs have insufficient background knowledge about circuits and those where incorrect reasoning occurs? Do errors caused by insufficient knowledge fall under Direction Errors?
2. A minor question about Table 4: how is the error rate calculated? I saw that the sum of each column or raw do not equal 1.

---

> ### Author Response · Authors · 2024-11-20
> **Clarifications on Error Types and Rates and Dataset Design and Imbalance**
>
> Thank you for your thoughtful comments and questions. We address your points as follows:
>
> **Questions:**
> 1. **Differentiating errors:**
> LLMs have not demonstrated a clear lack of background knowledge about circuits in our dataset, so we have not differentiated between the two types of errors in our evaluation. Errors are categorized as math, formatting, or reasoning errors. Topology errors are a subtype of reasoning errors, and direction errors are a subtype of topology errors. Most reasoning errors are topology errors; LLMs display sufficient background knowledge but struggle with its application. For further details on error types, please see Appendix D. Additionally, examples of reasoning errors (including topology and direction errors) can be found in Appendices F.1, F.3, F.4, and F.7.
>
> 2. **Error rate calculation:**
> Error rates are calculated as the ratio of data points with a specific error type to the total data points in the subset. For example, for questions with netlists, the subset consists of 79 templates × 5 setups = 395 data points. Each question is analyzed for all error types, and a single question may fall into multiple categories. For example, a question with a math and a direction error is counted under math, reasoning, topology, and direction errors. Thus, the percentages in Table 4 do not sum to 100% because errors can overlap. Constraints are: math, format, reasoning ≤ 100%, topology ≤ reasoning, and direction ≤ topology.
>
> **Weaknesses:**
> 1. **Sources of problems:**
> We agree that incorporating modern challenges is vital for broader analysis. Our dataset’s design allows for scalability to include advanced tasks, while textbook-derived problems provide a well-defined and reliable foundation for this initial work. Initial results show that LLMs struggle even with basic topology understanding, highlighting the need to first address these fundamental challenges before moving to more challenging circuits. Both are exciting directions for future work, and we shall highlight that in the revised discussion.
>
> 2. **Category and level imbalance:**
> We acknowledge the imbalance and agree that additional analysis would be useful. Table 5 in Appendix E provides category accuracies. In the revision, we can include a discussion of these results and an analogous table of per-level accuracies:
>
> | Dataset Subset                  | Prompt            |     |  GPT 4o Accuracy (%) per level       |       |
> |------------------------|-------------------|:-------------:|:-------------:|:-------------:|
> |                        |                  |       1       |       3       |       5       |
> | Questions With Netlists| 0-s w/ netlists  |   **49.4**    |     **31.2**      |     **18.5**      |
> |                        | 1-s w/ netlists  |     45.0      |     28.2      |      9.2      |
> | Questions W/O Netlists | 1-s              |     85.0      |     60.0      |     48.0      |
>
> Due to our level definition, we expect a high correlation between levels and categories. Addressing the challenge of achieving more balanced levels and category diversity is an exciting direction for future work.
>
> We hope this clarifies your questions and addresses your concerns. If you have any further questions or feedback, we would be happy to address them.

---

> > ### Comment · Reviewer_5hrA · 2024-11-29
> >
> > Thanks for authors in addressing my concerns. This is a valuable contribution to evaluating LLMs in the hardware domain. It would be beneficial to consider incorporating more diverse sources into this benchmark in future.

---

> > > ### Author Response · Authors · 2024-11-29
> > > **Acknowledging Feedback and Highlighting Key Revisions**
> > >
> > > Thank you for your thoughtful follow-up and for recognizing the value of our contribution. In the revision, we have clarified our dataset scope and limitations, added per-level accuracies alongside per-category results, discussed these results, and emphasized the dataset’s scalability,  which we are excited to leverage to further build on this work and enhance its impact. We would be grateful if you could take a moment to review the revised version, and we welcome any further questions or feedback. Thank you again for your valuable input!

---

### Official Review · Reviewer_JapT · 2024-11-03

**Soundness:** 1
**Presentation:** 2
**Contribution:** 1
**Rating:** 5
**Confidence:** 2

**Summary:**

This paper introduces a CIRCUIT dataset. The dataset contains 510 questions and is evaluated via three LLM based models.

**Strengths:**

* Evaluation results looks extensive, with results analysed in detail to have a comprehensive understanding of their performane on the dataset introduced.

**Weaknesses:**

* The dataset size may not be sufficient enough to fully evaliate the capabilities of LLMs in analogue circuit design.
* While the evaluation results do show there is imrovements in terms of accuracies on analogue circuit related questions, it is unclear how well does this translate in designing analogue circuits.
* The paper only tests on solutions using LLM only, it is not clear how circuit design using combination of different models could utilize this as a benchmark. This seems more like how to train a LLM using questions on how analogue circuits operate.

**Questions:**

please see weakness

---

> ### Author Response · Authors · 2024-11-20
> **Clarifications on Dataset Scope, Relevance, and Benchmark Utility**
>
> Thank you for your thoughtful comments and questions. We address your points as follows:
>
> **Dataset size:**
> We acknowledge the dataset's limited size but emphasize that it serves as a carefully curated starting point for evaluating LLMs in analog circuit reasoning. Importantly, the dataset is designed to be scalable, enabling future expansion to address more complex tasks. Initial results show that LLMs struggle even with basic topology understanding, highlighting the need to address these fundamental challenges before moving to advanced analog design tasks.
>
> **Relevance to analog circuit design:**
> Our benchmark focuses on interpreting and understanding circuit topologies, which are foundational to the analog design process. This task represents an essential building block for more complex design stages. LLMs seem to struggle with this basic task, and we believe that addressing this limitation is a necessary first step before integrating LLMs into the analog design process, as their usefulness and versatility in performing more complex tasks would be limited and minimal.
>
> **Benchmark Utility:**
> Our benchmark design is flexible and scalable, making it suitable for evaluating hybrid approaches that combine LLMs with other models or tools. The unit-test-like methodology ensures that if a model passes all setups for a given template, it reliably demonstrates an understanding of the topology, regardless of the specific reasoning process. This approach offers a consistent and objective evaluation framework that can be extended to hybrid models by integrating their outputs into the evaluation pipeline. We acknowledge the limitation that automatic evaluation focuses on the final answer and may overlook errors in intermediate reasoning, such as hallucinations or mathematical inaccuracies that do not propagate to the final output. Future work could include automatic intermediate-step evaluation strategies to address this limitation.
>
> We hope this clarifies your questions and addresses your concerns. If you have any further questions or feedback, we would be happy to address them.

---

> > ### Comment · Reviewer_JapT · 2024-11-26
> > **Response to Rebuttal**
> >
> > I thank the authors for responding to my questions on the paper and explaining these in detail.

---

> > > ### Author Response · Authors · 2024-11-29
> > > **Encouraging Feedback on Revised Work**
> > >
> > > We truly appreciate your engagement and the feedback that has helped us improve our work. In the revision, we have implemented several changes to enhance our paper's clarity, scope, and utility. Additionally, we moved some results from the appendix into the main paper, included additional results, and added their discussion to strengthen our contributions. We would be grateful if you could take a moment to review the revised version, and we welcome any further questions or feedback. Thank you again for your constructive input and thoughtful consideration.

---

### Author Response · Authors · 2024-11-27
**Revisions in Response to Reviewer Feedback**

We took the reviewer feedback and implemented several changes to our work. The main changes are highlighted in red. These include:

1. **Clarified the benchmark scope** and how it helps advance LLMs in analog circuit design in the Introduction and Related Work sections.
2. **Moved per-category accuracies** from the Appendix to the main paper body, **added a table of per-level accuracies**, and briefly discussed these results in the Human Evaluation section.
3. **Reorganized the Human Evaluation section** slightly and cut content for clarity.
4. **Emphasized the technical contributions** of our work in the Discussion section and reflected in more detail on future work. Also, slightly reorganized and cut content for coherence.
5. **Revised the Limitations** of our work based on reviewer feedback.
6. **Refined the Conclusion** for improved coherence.

We appreciate the reviewers' feedback, which helped us improve and clarify our contributions.

---

### Meta-Review · Area_Chair_JyzE · 2024-12-22

**Metareview:**

The paper describes a benchmark for understanding analog circuits, which consists of 510 questions. Different LLMs are evaluated. The setup requires numbers (like the voltage) and the diagram of the circuit in the netlist format, as well as the question/answer pairs.

The accuracy of the models like GPT-4 is around 50%

The usefulness of creating new benchmarks for LLM very clear, and this paper adds one of them. However, I think that the contribution is not so big: you can expect such kind of results for any university-level course, be it topology, algebraic geometry or circuit understanding: they all can be transformed into such microbenchmarks.
The paper does not provide any recommendations or attempts to increase the accuracy.
What happens, if we finetune the model on such circuit data? What are typical errors and in which part?
The benchmark also does not show any fundamental limitations of LLM for this scenario.

**Additional Comments On Reviewer Discussion:**

The reviewers agree that the paper is below the bar. Some of the concerns raised are the limitations of the design and experiments.
An interesting comment was that it is unclear if the benchmark could assist in designing new circuits (which could be of great practical value).

---

### Decision · Program_Chairs · 2025-01-22

Reject